# Menagerie: A text-mining tool to support animal-human translation in neurodegeneration research

Caroline J. Zeiss[1]*, Dongwook Shin[2], Brent Vander Wyk[3], Amanda P. Beck[4], Natalie Zatz[5], Charles A. Sneiderman[2], Halil Kilicoglu[2]

**1** Department of Comparative Medicine, Yale School of Medicine, New Haven, Connecticut, United States of America, **2** Lister Hill National Center for Biomedical Communications, National Library of Medicine, Bethesda, Maryland, United States of America, **3** Department of Internal Medicine, Yale School of Medicine, New Haven, Connecticut, United States of America, **4** Department of Pathology, Albert Einstein College of Medicine, New York, United States of America, **5** Department of Ecology and Evolutionary Biology, Yale University, New Haven, Connecticut, United States of America

* caroline.zeiss@yale.edu

**Data Availability Statement:** All relevant data are within the manuscript and its Supporting Information files.

## Abstract

Discovery studies in animals constitute a cornerstone of biomedical research, but suffer from lack of generalizability to human populations. We propose that large-scale interrogation of these data could reveal patterns of animal use that could narrow the translational divide. We describe a text-mining approach that extracts translationally useful data from PubMed abstracts. These comprise six modules: species, model, genes, interventions/disease modifiers, overall outcome and functional outcome measures. Existing National Library of Medicine natural language processing tools (SemRep, GNormPlus and the Chemical annotator) underpin the program and are further augmented by various rules, term lists, and machine learning models. Evaluation of the program using a 98-abstract test set achieved $F_1$ scores ranging from 0.75–0.95 across all modules, and exceeded $F_1$ scores obtained from comparable baseline programs. Next, the program was applied to a larger 14,481 abstract data set (2008–2017). Expected and previously identified patterns of species and model use for the field were obtained. As previously noted, the majority of studies reported promising outcomes. Longitudinal patterns of intervention type or gene mentions were demonstrated, and patterns of animal model use characteristic of the Parkinson's disease field were confirmed. The primary function of the program is to overcome low external validity of animal model systems by aggregating evidence across a diversity of models that capture different aspects of a multifaceted cellular process. Some aspects of the tool are generalizable, whereas others are field-specific. In the initial version presented here, we demonstrate proof of concept within a single disease area, Parkinson's disease. However, the program can be expanded in modular fashion to support a wider range of neurodegenerative diseases.

**Funding:** CJZ was supported by a fellowship program at the U.S. National Library of Medicine, National Institutes of Health. HK, DS, and CS were supported by the intramural research program at the U.S. National Library of Medicine, National Institutes of Health. BV was supported by the National Institute on Aging Yale Claude D. Pepper Older Americans Independence Center [P30AG021342]. The funders had no role in study design, data collection and analysis, decision to publish, or preparation of the manuscript.

**Competing interests:** The authors have declared that no competing interests exist.

# Introduction

Despite high rates of success reported in animal studies, many promising interventions for neurodegenerative and other complex diseases do not translate to effective therapies in humans [1–3]. Reasons for this translational gap are complex, [4] and occur at all stages of the preclinical [5, 6] and clinical [7, 8] continuum. In clinical trials, failed efficacy remains a primary translational roadblock, particularly in phase II and III trials [7, 9]. In contrast to this reality, discovery literature is heavily weighted toward publication of promising studies in animals [10–13]. In the area of neurodegeneration, this translational gap has fueled skepticism regarding the validity [3, 14, 15] and cost [16] of preclinical animal studies.

Several factors that undermine the reliability of animal studies have been identified. These include insufficient rigor in animal study design and reporting [6, 17], publication bias [10, 12], over-reporting of significance [18, 19], over-reliance on non-clinical outcome measures [20], and entrenched use of certain model systems [3, 21, 22]. Together, these issues contribute to poor reproducibility of animal studies, and certainly worsen the translational gap [23]. To address this, reporting and design guidelines [24] have been adopted by numerous journals [25], and by major funding agencies [26]. These, once implemented, should improve rigor and reproducibility of preclinical studies, although widespread evidence for this is not yet available [27]. Regarding publication bias, greater reporting of negative studies [28, 29] would provide a more realistic view of actual preclinical efficacy.

Whether these changes alone will improve translatability is unclear. In the case of neurodegenerative disease, the biological complexity of the research problem is a major hurdle. To understand mechanistic phenomena that may be obscured by this complexity, the prevailing approach is to apply hypothesis-based and reductionist methodology to genetically altered animal systems [30]. This has allowed insights that would not have otherwise been possible. However, a drawback of these systems is that their approaches are so specific that their results do not to generalize to other more complex situations i.e. there is limited external validity [2]. The challenges of extending findings from reductionist model systems to patient populations [2, 30–32] are immense. Profound differences in animal and human physiology [33] are a critical source of poor translation. Additionally, study design choices that influence generalizability extend beyond those needed to ensure unbiased study design. These are more contextually defined and concern the relationship between variables such as model choice [34] and mechanism of the intervention [22], integration of biomarker data with clinically relevant outcome measures, and use of progressive disease models and longitudinal study designs if neuroprotection is claimed. Addressing the potential generalizability of interventional animal studies is the impetus for the methods described in this paper.

PubMed (https://www.ncbi.nlm.nih.gov/pubmed/) is the largest global public repository of biomedical literature [35]. It is a rich source of animal model discovery data that, if harnessed on a large scale using automated methods, could conceivably be used to inform translational potential of a given therapeutic mechanism or approach. Many examples of automated methods to more effectively search [36], curate [37] or generate new knowledge using literature-based discovery methods [38] from this resource have been described. However, none focuses specifically on the issue of animal-human translation, while accommodating the unfortunate realities listed above that undermine the reliability of published data, even in prestigious journals [39]. In this paper, we describe a text-mining approach that aggregates abstract level data to support subsequent manual evaluation of the generalizability, or external validity, of animal studies in an area that is particularly difficult to model effectively—neurodegeneration. Because we cannot recapitulate human neurodegenerative diseases with any single animal model [40], an alternate approach may be to aggregate evidence across a diversity of models

[41] that capture different aspects of a multifaceted cellular process. Consistent results across such a diversity of systems may clarify whether an approach has translational potential [42, 43], particularly if these results extend across comparable clinically relevant outcome measures.

In the initial version presented here, we describe the program design and its capacity to accurately extract data of potential translational relevance within a single disease area, Parkinson's disease (PD). Animal models have been instrumental to development of symptomatic therapies for PD [44, 45]. However, PD confronts the same roadblocks as other neurodegenerative diseases in development of disease altering therapies [46, 47]. We demonstrate retrieval of the expected patterns of animal species and model selection [48, 49] for PD, and confirm previously identified [22] patterns of animal model use. This study paves the way for large scale evaluation of the entire PD corpus to identify those therapeutic approaches that have potential for clinical translation. Components of the tool are modular and readily adaptable to other neurodegenerative disease areas such as Alzheimer's disease.

## Materials and methods

We used a text mining approach to extract characteristics that we have previously determined, using manual curation, to be reliably present in abstracts, and to be translationally useful [22, 50]. These comprise information regarding therapeutic intervention, molecular target (s) or genes, species, model, overall outcome of the study and whether functionally relevant outcome measures [20] were reported. A dataset of 504 PubMed abstracts was manually annotated to develop/refine our approach and to validate it. To assess the utility of the approach, we also applied it to a larger dataset of 14,481 Parkinson's disease abstracts. Below, we first present the data collection process. Next, the text mining components are described. We conclude this section by describing the validation of the approach on the manually annotated dataset, and on the 14,481abstract dataset. A pictorial summary of our approach is provided (Fig 1).

### 1. Data collection

**a. Evaluation dataset.** A total of 504 PubMed abstracts were used to develop and evaluate the text mining components. This dataset consists of two parts. The training set (406 abstracts), developed in an earlier study [22], spans years 2015–2016 and was used for development and

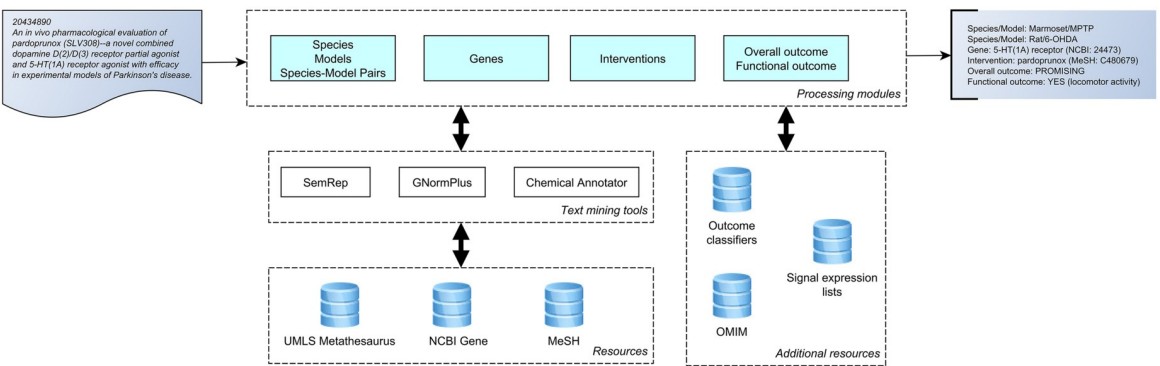

**Fig 1. A system overview of the text-mining tool.** Modules (species, models, genes, interventions and outcomes) defining types of data returned (far right) from an abstract (far left) are shown on the first row. This is achieved through a combination of rule-based and machine learning methods utilizing established terminological resources (Unified Medical Language System (UMLS) Metathesaurus, NCBI Gene and Medical Subject Headings (MeSH)), additional text mining tools (SemRep, GNormPlus and the chemical annotator) and series of signal lists and rules (outcome classifiers and signal expression lists supplemented from databases such as OMIM).

refinement of our text mining components. The test set (98 abstracts), collected for this study, was drawn from year 2017 and was used to validate the components. Criteria used to select PubMed IDs (PMIDs) have been previously described [22]. Only interventional studies were included in training and test sets—these were identified by manual evaluation of the title and were defined as those in which the effect of an intervention (pharmaceutical, phytochemical, physical, genetic, behavioral or environmental) on the PD phenotype was examined.

The annotation process was carried out using primarily Excel spreadsheets. Brat annotation tool [51] was used to verify intervention and gene annotations, which tended to be more complex. The annotation process differed in some respects between the characteristics considered:

- Species and model annotations were carried out in tandem at the abstract level. In addition to individual species and models relevant to the study (e.g., Mouse and MPTP, respectively), species/model combinations that were reported in the abstract were also annotated (e.g., Mouse/MPTP).

- Gene names were annotated at the mention level; that is, all occurrences of these terms in the title and abstract were annotated. They were also mapped to identifiers in the NCBI Gene database (https://www.ncbi.nlm.nih.gov/gene) and to concept identifiers in the UMLS Metathesaurus (https://uts.nlm.nih.gov/metathesaurus.html). For example, both *dopaminergic D1 receptor* and *D1 receptor* were mapped to NCBI Gene identifier 24316.

- Interventions/disease modifiers were annotated at the abstract level. For meaningful evaluation, they were also mapped to standard identifiers in the MeSH vocabulary (https://meshb.nlm.nih.gov/search), the NCBI Gene database, the UMLS Metathesaurus. For example, both *selegiline* and *deprenyl* were mapped to the MeSH descriptor D012642.

- Overall outcome was annotated at the abstract level, and the outcome value (promising, negative, mixed, and other) was determined based on the title and the last two sentences of the abstract. These terms are defined more fully in the section below. While individual outcome measures (e.g., *dyskinesia*), polarity-signaling expressions (e.g. *reduced*), and sentence-level outcomes were annotated in the training set, these were ultimately not used. We did not perform specific annotation for functional outcomes, though a list of functional outcome terms was compiled from the outcome measures annotated in the training set. In the test set, only overall outcome was annotated. Two of the authors (CJZ and CS) annotated the overall outcome in 54 abstracts independently to establish guidelines for annotating this element. The interannotator agreement was found to be 0.55 (Cohen's $\kappa$, moderate agreement). CJZ later adjudicated these annotations and annotated the rest of the abstracts.

**b. Text mining components.** Text mining components were developed in Java programming language. Existing natural language processing (NLP) tools underpin the components and are further augmented by various rules, term lists, and machine learning models. NLP tools used are the following:

- *SemRep* [52] is a rule-based system that maps text in PubMed abstracts to UMLS Metathesaurus concepts with broad semantic classes, such as Pharmacologic Substance or Mammal (and relationships between concepts, which were not used in this study).

- *GNormPlus* [53] is a machine learning-based system that identifies gene/protein terms in text and maps them to NCBI Gene identifiers.

- *Chemical annotator* is a dictionary-based method developed at the U.S. National Library of Medicine that extracts chemical entities from text and normalizes them to MeSH identifiers.

We implemented our approach as six individual modules: a) species, b) model, c) genes, d) interventions/disease modifiers, e) overall outcome and f) functional outcomes. Each module extracts relevant information from a list of abstracts and the results are stored in a relational database (MySQL) for various types of analyses. Details of each module are provided below.

**Species**: Humans and animal species considered in this study included those most commonly used in research: non-human primates, rodents, larger less commonly used mammals (e.g., dog, cat, rabbit) and a selection of commonly used non-mammalian species (e.g., fly, worm, frog and fish). To identify these species, we used rules that map UMLS concepts identified by SemRep and other signal expressions to species terms. For example, the UMLS concept C0008976:Clinical Trials was mapped to species Human and C0006764:Callithrix to Marmoset. These rules account for synonyms describing the same species (e.g., *cat* and *feline* for Cat), mapping them to a single species term to facilitate subsequent analysis. Terms identified at the sentence level were consolidated to summarize the spectrum of species mentions at the abstract level. Two rules were used to exclude species terms that, while mentioned in the abstract, may not be relevant to the study under examination:

- Species terms identified only in the background section of the abstract (if structured) or in the first sentence of the abstract (if unstructured) were excluded. We found this approach was needed to avoid extraneous collection of species terms used in background text summarizing previous findings across species.

- Human was excluded as species if it was found only in the conclusion section of the abstract (if structured) or within the final 15% of the sentences of the abstract (if unstructured). This was done to ensure that abstracts for animal studies that included discussion about potential in human populations were assigned as animal studies, not human studies.

Additionally, we established a hierarchy of terms so that those referring to specific species (e.g., Macaque) were returned in favor of more general terms (e.g., Non-human primate) occurring in the same abstract. In the event that no specific species terms (e.g., Macaque) were used in the abstract, more general terms can be extracted. Rules to collect species data are generalizable to any area of study.

**Model**: In contrast, rules used to collect model data were partly Parkinson's disease-specific. The methodology used for models was largely the same as that used for species. A series of signal expressions were used to capture commonly used models (e.g., 1-methyl-4-phenyl-1,2,3,6 tetrahydropyridine → MPTP and 6-hydroxydopamine → 6-OHDA, and others) that are largely specific to PD. In addition (and generalizable to any disease area), signal expressions were included to identify the category of genetically altered animal models. These can capture genetically altered models regardless of disease area, and were thus PD-independent. These signal expressions also identified human studies of patients with mutations in PD associated genes. As for species, model was identified at the sentence level, but consolidated at the abstract level to provide a list of the spectrum of model mentions as the final readout. Species and models were identified in tandem. If a species term and a model term are identified within a pre-specified window of each other in the text (3 phrases), they were also paired at the abstract level to make the species/model link explicit.

**Genes**: When considered together with data collected from other modules, gene data allows the user to assess the extent to which molecular pathways or potential targets associated with a given intervention are preserved across model systems. Extraction of gene terms is underpinned by GNormPlus [53] and SemRep [52]. SemRep results were filtered to concepts with one of three UMLS semantic types: Gene or Genome; Amino Acid, Peptide, or Protein; and Enzyme. We augmented GNormPlus and SemRep with a set of gene names with mutations/

polymorphisms known to be associated with PD in the Online Mendelian Inheritance in Man (OMIM) database. This dictionary contains 28 such gene names and 818 synonyms associated with them (e.g., *parkin* for PRKN and *alcohol dehydrogenase 3* for ADH1C). This list was designed to address potential misses by GNormPlus and SemRep, that can be considered essential for an automated tool to extract. The module searches the abstract for the presence of synonyms and records the gene name associated with the synonym, if found. This list can be supplemented and updated by code that can extract a list for a given disease from OMIM.

Additional rules were used to address several kinds of false positive errors that were identified during the training process. These rules filter out ordinal numbers (e.g., *2nd*), confidence intervals (e.g., *CI 0.95*), gene names that map to common English words (e.g., *all*, *impact*), non-specific genetic terms (e.g., *transcription factor*, *protein*, *candidate gene*), and terms relevant to other modules (e.g., *dopa*, *mptp(+)*, and *liraglutide*). Other terms were excluded because they were not annotated in the training set, even though they seemed legitimate from an extraction point of view (e.g., *neurotrophin*, *glutamate*, *acetylcysteine*). To facilitate subsequent analysis of returned data, NCBI gene identifiers (for gene terms returned using GNormPlus or the OMIM dictionary) or CUIs (for gene terms returned using SemRep) were also collected. In this way, synonyms associated with the most current gene nomenclature mapped to a single identifier.

**Interventions/Disease Modifiers**: This module aims to capture pharmaceutical, phytochemical, physical, genetic, behavioral or environmental entities that could alter a disease phenotype. The chemical annotator, in addition to SemRep and GNormPlus, provided the basis for this module, augmented with a list of essential interventions. To address the spurious entities identified with the NLP tools, we also use a list of exclusion terms. The chemical annotator maps identified interventions to MeSH identifiers. UMLS concepts identified with SemRep were filtered based on their semantic types. The filter included the following types: Amino Acid, Peptide, or Protein; Biologically Active Substance; Chemical; Food; Hazardous or Poisonous Substance; Hormone; Inorganic Chemical; Organic Chemical; Substance; and Vitamin. GNormPlus was used to identify genetic interventions from the titles only. The list of essential terms was identified in manually annotated sets from two previous publications [22, 54] and was supplemented by interventions given in Alzforum therapeutics (https://www.alzforum.org/therapeutics; searched September 10, 2018) and the Michael J Fox Foundation (https://www.michaeljfox.org; searched September 10, 2018). Interventions for both PD and Alzheimer's disease (AD) were included, recognizing that some interventions (particularly for neuropsychiatric complications) are shared across different neurodegenerative diseases [55]. The exclusion list was based on our observations on the training set and included generic terms like "treatments". Database identifiers were used to account for synonyms and consolidate sentence-level terms to summarize the spectrum of interventions identified at the abstract level. Gene mentions in the title only are extracted in this module. Because many genes are mentioned in the body of the abstract, using the title only this enriches data for those molecular entities that are thought to influence the PD phenotype significantly and are potential therapeutic targets.

**Overall outcome**: Our intent with this module was to distill the conclusion of the study (as defined by the authors) regarding the overall potential of the study to influence trajectory of the disease. Four final categories were created: those in which outcomes held therapeutic promise (PROMISING), those with adverse effects (NEGATIVE), those with both promising and adverse effects (MIXED) and those in which outcomes were indeterminate (OTHER). This module is implemented as an ensemble of machine learning models. Specifically, support vector machine (SVM) models were developed for positive and negative outcomes. The positive outcome model classified each abstract as POSITIVE or NOT-POSITIVE. The negative outcome model

classified each abstract as NEGATIVE or NOT- NEGATIVE. The final decision on overall outcome of a study was made based on the predictions of the two models:

- POSITIVE + NOT-NEGATIVE → PROMISING

- POSITIVE + NEGATIVE → MIXED

- NOT-POSITIVE + NEGATIVE→ NEGATIVE

- NOT-POSITIVE + NOT-NEGATIVE→ OTHER

Features extracted from the title and the last two sentences (the context) were used for classification. The features were adapted from Niu *et al.*[56], who used a classification approach to determine polarity of clinical outcomes (no outcome, positive, negative, neutral outcome). These features are the following:

- *n-grams*: unigrams and bigrams from the context, stemmed using Porter stemmer [57].

- *Change word features*: Two sets of words are defined for these features: MORE (15 terms, e.g., *increase*) and LESS (34 terms, e.g., *alleviate*). The tag MORE is added to all words between a MORE word and the next punctuation, and the tag LESS to the words after a LESS word, aiming to capture the scope of these words.

- *Change/polarity word co-occurrence*: These capture co-occurrence of *change words* and *polarity words*, which can indicate positive/negative assessment. To extract these features, two additional set of terms were created for GOOD (49 terms, e.g., *ameliorate*) and BAD (12 terms, e.g., *exacerbate*) terms. 4 features were generated by combining the four classes: MORE GOOD, MORE BAD, LESS GOOD, LESS BAD, and if terms from respective categories appeared within a pre-specified window (4 words), that feature was set to 1. We also used terms that are categorized as Disorders in the UMLS semantic groups [58] as BAD terms. Therefore, a sentence with the fragment *alleviate Parkinson's disease* would be encoded by setting the LESS BAD feature to 1.

- *Negation*: All words modified by the negation *no* in a sentence are appended with NEG, and used as additional features (e.g., *no significant change* → {significant_NEG, change_NEG}).

- *Semantic type*: We encode each UMLS semantic type as a feature and set this feature to 1, if a concept with a given semantic type exists in the context.

LIBLINEAR implementation of linear SVM [59] was used for training the models.

**Functional outcomes**: Functional outcomes were defined as those that reflected the effect of an intervention on measurable variables in a living organism (e.g., physical movement or survival). A set of such outcome terms was collected manually from the training set. These include terms common in human clinical trials (e.g., *the Unified Parkinson's Disease Rating Scale*), general terms reflecting neurologic function (e.g., *bradykinesia*) and terms specific to animal model studies (e.g., *rotarod performance* or *turning behavior*). We then simply checked whether an abstract contained any of these terms, and categorized the abstract as YES, if it did, and as NO, if no functional outcome measure was detected.

**Additional data**: We collected metadata related to abstracts from PubMed records for subsequent analysis. These included year of publication, journal, and publication type.

## 2. Evaluation

**a. Intrinsic evaluation.**   After developing and refining the modules using the training data, we validated them on the test set (98 abstracts). We used standard information extraction

evaluation metrics: precision (or positive predictive value), recall (sensitivity), and $F_1$ score (the harmonic mean of precision or recall).

All modules were evaluated at the abstract level. Gene/protein and intervention modules were also evaluated at the mention level. For the abstract level evaluation, multiple extractions of the same term are consolidated into a single extraction (i.e., only unique terms are considered). For the mention level evaluation, all instances of a term are considered separately. Additionally, the intervention module was evaluated based on title only (i.e., only interventions extracted from the title and the title annotations were compared). For evaluating gene and intervention modules, we also used both exact matching and approximate matching. In exact matching (stricter evaluation), a term extracted by the module should exactly match an annotated term with respect to term boundaries to count as a true positive. In approximate matching, overlap of the terms is acceptable.

Given that the annotated dataset is relatively small (504 examples for overall outcomes) and it is standard to evaluate machine learning models that are trained on small datasets using cross-validation, we evaluated the overall outcome module using 10-fold cross-validation on the full dataset, instead of using the training-test split. We repeated the cross-validation experiment 50 times and reported the average of results. Overall outcome models that were used in the large scale evaluation (below) were trained on the full dataset. We also compared evaluation results from our program, where applicable, with results from comparable programs applied to the same 98-abstract test set. These baseline systems include PubTator [18], GNormPlus [44], and SemRep [43] with semantic filtering.

**b. Large scale evaluation.** To assess the utility of our approach on a larger scale, we collected a dataset of 14,481 abstracts using "Parkinson's disease" as the sole search criterion in PubMed (searched 11/14/2017). These spanned three time points across 10 years (2008; 3433 PMIDs, 2012; 4727 PMIDs and 2017; 6321 PMIDs; S1 File). In contrast to the evaluation dataset, these abstracts were not enriched for interventional studies by subsequent manual selection of studies with titles implying use of an intervention. This allowed us to assess utility of the program to extract interventional studies from the broadest search possible. Data was analyzed manually using Excel or using a series of queries to aggregate studies by their features (species, model, intervention, outcome, functional outcome, and genes) within and across publication years. Specific queries that implemented logical conjunctions of feature sets (e.g. Species = 'Mouse' AND outcome = 'PROMISING') were used to generate lists of publications that met the criteria. For each query the list of qualifying publications was outputted along with a set of summary statistics such as the count and proportion of publications per year to facilitate further investigation and data visualization. Using standardized query methods affords the opportunity to specify arbitrarily complex queries to support user-driven exploration of the database. Specific comparisons are given by subheading in the results section.

**Data access**: All supporting data is available as Supplementary Data or on request from CJZ or HK.

## Results

### 1. Intrinsic evaluation of the program

Precision, recall and $F_1$ scores for each module were assessed using exact matching (all modules) as well as approximate matching (Genes and Interventions modules) at the title, abstract and mention level as shown in Table 1.

**Species and model modules.** Using exact matching, we obtained higher recall compared to precision; results were similar for species and models. The performance for species/model

**Table 1. Results of program evaluation on a 98-abstract test set.**

| Module | Precision | Recall | $F_1$ | Accuracy |
|---|---|---|---|---|
| **A. Species and model module** | | | | |
| Species (exact matching) | 0.84 | 0.90 | 0.87 | - |
| Model (exact matching) | 0.87 | 0.91 | 0.89 | - |
| Species/Model (exact matching) | 0.72 | 0.79 | 0.75 | - |
| **B. Genes module** | | | | |
| Mention level (approximate matching) | 0.88 | 0.85 | 0.87 | - |
| Mention level (exact matching) | 0.79 | 0.77 | 0.78 | - |
| Abstract level (approximate) | 0.80 | 0.83 | 0.82 | - |
| Abstract level (exact) | 0.74 | 0.76 | 0.75 | - |
| **C. Interventions/disease modifiers module** | | | | |
| Abstract level (approximate matching) | 0.96 | 0.43 | 0.59 | - |
| Abstract level (exact matching) | 0.85 | 0.37 | 0.52 | - |
| Title only (approximate) | 0.95 | 0.83 | 0.89 | - |
| Title only (exact) | 0.83 | 0.73 | 0.78 | - |
| **D. Overall outcome evaluation (10-fold cross-validation)** | | | | |
| Promising | 0.91 | 1.0 | 0.95 | - |
| Negative | 0.96 | 0.70 | 0.81 | - |
| Mixed | 0.86 | 0.59 | 0.70 | - |
| Other | 0.79 | 0.53 | 0.64 | - |
| | | | | 0.90 |
| **E. Functional outcome measure** | | | | |
| Functional outcome (yes) | 0.87 | 0.90 | 0.89 | - |
| | | | | 0.86 |

combination was lower, as expected, since both the species and model needed to be correct and the pre-specified window size (3 phrases) can cause additional errors.

**Genes module**: We obtained similar precision and recall with the genes module. When approximate matching was used for evaluation, the results were better both at the abstract and the mention level, indicating that identifying precise gene/protein term boundaries in text is a challenge.

**Intervention/disease modifier module.** The results were obtained by considering interventions extracted from the title only using semantic type filtering. Performance at the abstract level was significantly lower, because even though the evaluation was performed at the abstract level, the module still extracted interventions only from the title. The difference between the performance of the system at the abstract level and title only evaluation (0.52 vs. 0.78 $F_1$ score with exact matching) indicates that a significant number of terms defined in our program as interventions/disease modifiers were discussed in the abstract only.

**Overall outcome and functional outcome measure.** The results of 10-fold cross-validation for overall outcome prediction are provided in Table 1D. Note that the results are the mean average of 50 cross-validation experiments. In contrast to overall outcome, functional outcome (yes/no) results were based on the test set, since the functional outcome term list was derived from the training set.

Next, we compared the results of our program to those achieved by comparable baseline programs PubTator [37], GNormPlus [53] and SemRep [43] with UMLS semantic type filtering (Table 2).

**Table 2. Comparison with evaluation results from comparable baseline programs (98-abstract test set).**

| Module | Precision | Recall | F$_1$ | Accuracy |
|---|---|---|---|---|
| **A. Species and model module** | | | | |
| Species (Menagerie; exact matching) | 0.84 | 0.90 | 0.87 | - |
| Species (PubTator) | 0.74 | 0.66 | 0.70 | - |
| **B. Genes module** | | | | |
| Mention level (Menagerie; exact matching) | 0.79 | 0.77 | 0.78 | - |
| Mention level (GNormPlus; exact matching) | 0.93 | 0.54 | 0.69 | - |
| **C. Interventions/disease modifiers module** | | | | |
| Abstract level (Menagerie; exact matching) | 0.85 | 0.37 | 0.52 | - |
| Abstract level (Chemical and Substances in UMLS; exact matching) | 0.48 | 0.26 | 0.34 | - |
| **D. Overall outcome evaluation** | | | | |
| Promising (Menagerie; 10-fold cross-validation) | 0.91 | 1.0 | 0.95 | - |
| Overall accuracy (Menagerie) | | | | 0.90 |
| Promising (majority vote) | 0.76 | 1.0 | 0.86 | - |
| Overall accuracy (majority vote) | | | | 0.76 |
| **E. Functional outcome measure** | | | | |
| Functional outcome (Menagerie; yes) | 0.87 | 0.90 | 0.89 | 0.86 |
| Functional outcome (Finding and Sign or Symptoms UMLS; yes) | 0.67 | 0.87 | 0.76 | 0.64 |

Our program achieved improved precision, recall and F1 scores for Species when compared to PubTator. For genes, GNormPlus output was used as-is, with exact matching. Our program achieved improved recall and F1 scores, at the cost of some precision loss. For interventions, concepts within the semantic group Chemicals and Drugs in UMLS were used- our program achieved improved values by all criteria. As our program utilizes annotated signal terms in addition to UMLS concepts, this was expected. For outcomes, a majority vote, which assigned PROMISING overall outcome to all abstracts, was used, and achieved lower performance than our program. For functional outcomes, concepts with the UMLS semantic types Finding and Sign or Symptoms were used as functional outcomes. Our program achieved significantly improved values by all criteria, again most likely due to annotated signal terms specifically directed at PD. Next, in the absence of a comparable baseline program for animal models or functional outcome measures, we assessed this aspect in a larger scale evaluation.

## 2. Large-scale evaluation results

The evaluation goals for this dataset were to assess those modules that could not be assessed using other programs (models, overall outcome and functional outcome measure) and to demonstrate how data could be aggregated to evaluate the diversity of systems across which a therapeutic approach or cellular mechanism had been assessed.

**a. Descriptive characteristics. Overall outcome by publication types**: Each abstract (defined by a unique PMID) was assigned primary or secondary data status on the basis of its associated publication type. In all three years, primary data constituted between 70% and 75% of all publications (S1 Table). Because publications are biased towards those reporting promising outcomes [10], we hypothesized that this tendency would be amplified by including secondary publication types in our comparisons. To test this hypothesis, we compared the proportions of outcome types in primary data sources alone, to those including both primary and secondary data sources. In general, approximately 70% of studies report a promising outcome with proportions in primary data sources being slightly lower than in all data sources

(S1 Table). In both datasets, a slightly increasing trend in promising reports across 10 years was noted. Primary data was used for subsequent comparisons.

**Species and animal model use over 10 years**: Overall, studies in humans prevailed (41%), followed by studies using mice and rats (approximately 10%), and non-human primates (mentioned in 1% or fewer of studies; Table 3). Predominance of rodents and use of non-human primates as experimental models in PD is well-established [41, 60]. Increasing trends of species mentions were strongest for zebrafish, drosophila and worm studies, reflecting increased use of lower species in mechanistic studies [61–63]. With the exception of marmosets, non-human primate use was flat or declining over time. 21% of studies over the 10-year period did not report species, while an approximately further 13% of studies mentioned animals in generic terms (animal model, rodent, non-human primate).

Model co-mentions for animals reported at the species level are given in Table 4. Rat and mouse models were by far the most heavily cited—of these the expected preference for (6-hydroxydopamine) 6-OHDA use in the rat and for (1-methyl-4-phenyl-1,2,3,6-tetrahydro-pyridine) MPTP use in mice and non-human primates was captured [48, 64]. Use of these two toxins and their variant applications, including hemi-parkinsonism, levodopa induced dyskinesia (LID) and 1-methyl-4-phenylpyridinium (MPP) toxicity constituted the majority of model choices. Chronic toxic models such as MPTP/probenecid, rotenone, paraquat, lacatcystin and maneb [64], traditional pharmacologic models (reserpine, haloperidol and galantamine) [65] and inflammatory models [66] were less common, but consistently used. As expected, mice [67] and lower species were the most heavily utilized genetically altered species.

Results confirm that the expected spectrum of vertebrate and invertebrate animal species and models used in PD research were captured [22, 68–71]. Additionally, the previously noted

**Table 3. Species mentions in Parkinson's disease research (2008–2017).**

|  | 2008–2017 | 2008 | 2012 | 2017 |
|---|---|---|---|---|
| Human | 4944 (0.41) | 1037 | 1641 | 2266 |
| Species not reported | 2615 (0.21) | 626 | 895 | 1094 |
| Animal model | 1336 (0.11) | 347 | 471 | 518 |
| Mouse | 1271 (0.10) | 261 | 439 | 571 |
| Rat | 1107 (0.09) | 313 | 373 | 421 |
| Rodent, unspecified | 187 (0.02) | 50 | 69 | 68 |
| Non-human primate | 155 (0.01) | 47 | 56 | 52 |
| Macaque | 70 (0.006) | 17 | 32 | 21 |
| Marmoset | 24 (0.002) | 8 | 4 | 12 |
| Vervet monkey | 5 (0.0004) | 4 | 0 | 1 |
| Baboon | 2(0.0002) | 2 | 0 | 0 |
| Cat | 38 (0.003) | 7 | 17 | 14 |
| Dog | 10 (0.0008) | 3 | 4 | 3 |
| Rabbit | 4 (0.0003) | 0 | 0 | 4 |
| Frog | 5 (0.0004) | 1 | 2 | 2 |
| Zebrafish | 30 (0.003) | 3 | 7 | 20 |
| Drosophila | 132 (0.01) | 27 | 48 | 57 |
| Worm | 61 (0.005) | 12 | 27 | 22 |
| Yeast | 64 (0.005) | 22 | 27 | 15 |
| **Total** | **12060** | **2787** | **4112** | **5161** |

Total numbers of publications (defined by unique PMID) are given by time period. Proportions are included in parentheses for the 2008–2017 period. Primary data only are used.

**Table 4. Species/model mentions in Parkinson's disease research (2008–2017).**

| Species and model | 2008 | 2012 | 2017 | Total PMIDs/ Species (% toxic models) |
|---|---|---|---|---|
| **Macaque** | | | | |
| MPTP | 7 | 13 | 7 | |
| Hemi-parkinsonian | 1 | 1 | 0 | 29 (100) |
| **Marmoset** | | | | |
| MPTP | 6 | 3 | 6 | |
| 1BnTIQ | 2 | 0 | 0 | 17 (88) |
| **Rat** | | | | |
| 6-OHDA | 93 | 128 | 119 | |
| Hemi-parkinsonian | 26 | 22 | 31 | |
| MPP | 9 | 7 | 5 | |
| Lipopolysaccharide induced model | 8 | 5 | 9 | |
| Genetically altered model | 6 | 6 | 9 | |
| Levodopa-induced dyskinesia | 6 | 10 | 10 | |
| Rotenone | 6 | 17 | 35 | |
| Haloperidol | 4 | 5 | 6 | |
| MPTP | 2 | 10 | 9 | |
| Reserpine | 2 | 5 | 2 | |
| Tremulous jaw movement model | 2 | 0 | 0 | |
| Galantamine | 1 | 0 | 0 | |
| Lactacystin | 1 | 0 | 3 | |
| Maneb | 1 | 0 | 0 | |
| Paraquat | 1 | 1 | 2 | 624 (78) |
| **Mouse** | | | | |
| MPTP | 69 | 98 | 138 | |
| Genetically altered model | 39 | 86 | 88 | |
| 6-OHDA | 10 | 20 | 36 | |
| Lipopolysaccharide induced model | 4 | 5 | 14 | |
| Paraquat | 4 | 5 | 4 | |
| Lactacystin | 3 | 1 | 3 | |
| Reserpine | 3 | 1 | 1 | |
| Rotenone | 2 | 2 | 15 | |
| Levodopa-induced dyskinesia | 1 | 1 | 5 | |
| Maneb | 1 | 3 | 2 | |
| MPP | 1 | 5 | 6 | |
| MPTPp | 1 | 0 | 1 | 678 (58) |
| **Zebrafish** | | | | |
| Genetically altered model | 1 | 0 | 1 | |
| MPTP | 1 | 0 | 2 | 5 (60) |
| **Drosophila** | | | | |
| Genetically altered model | 6 | 8 | 9 | |
| Paraquat | 1 | 1 | 4 | 29 (0) |
| **Worm** | | | | |
| Genetically altered model | 4 | 5 | 1 | |
| Rotenone | 1 | 0 | 0 | 11 (0) |
| **Yeast** | | | | |

*(Continued)*

**Table 4.** (Continued)

| Species and model | 2008 | 2012 | 2017 | Total PMIDs/ Species (% toxic models) |
|---|---|---|---|---|
| Genetically altered model | 2 | 1 | 0 | 3 (0) |

Models are listed by species in which they are mentioned, with 6-OHDA and MPTP models and their variants shaded in gray (column 1). Total numbers of publications in which models and species are co-mentioned in black in the last column. Of these, proportions of 6-OHDA and MPTP models and their variants (toxic models) are shown in parentheses.

dominance of 6-OHDA and MPTP-associated toxic models [22] was reiterated. These results confirmed that our tool identifies expected patterns of species and model use for the field. Next, we assessed its capacity to extract large-scale patterns regarding therapeutic interventions and gene mentions.

**Interventions/disease modifier mentions over 10 years**: 1862 unique entities (derived from UMLS or MeSH terms) were identified in the entire dataset. These were ranked by the number of associated publications at each time point, allowing identification of the most highly studied entities, as well as the trend of study over time (Fig 2, S2 Table).

Results are consistent with published data describing established treatments [72, 73] and recent approaches [74, 75].

**Gene mentions over 10 years**: This module extracts gene or molecular concept mentions in the entire abstract body, and thus identifies a larger group of genes and mechanisms than in the previous module (Fig 3, S3 Table). 3311 unique entities (derived from UMLS or NCBI gene terms) were identified in the entire dataset. Multiple synonyms describing the same entity mapped to a common UMLS or NCBI identifier. UMLS derived terms exhibit some redundancy with NCBI origin terms (e.g. dopamine transporter and SLC6A3) and capture groups of functionally related entities (e.g. D2 receptors) or molecular concepts (e.g. proteome). Proteins used as routine immunohistochemical markers or standard molecular reagents (e.g. tyrosine hydroxylase and GFAP) are captured. The expected dominance of alpha-synuclein was evident [76].

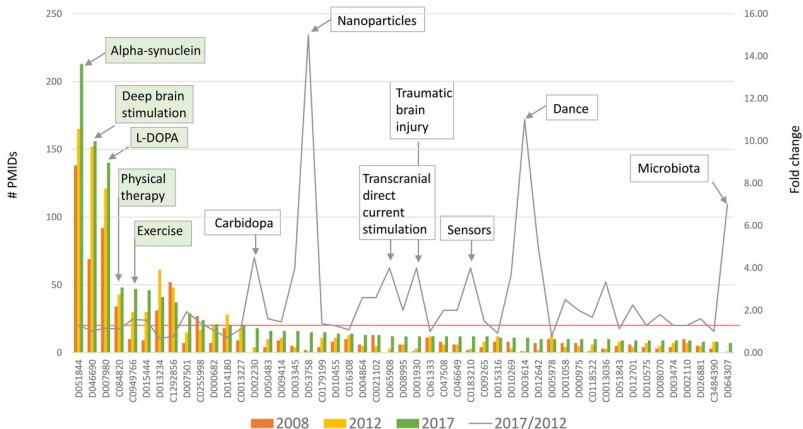

**Fig 2. Number of publications, by intervention or disease modifier (2008–2017).** Pharmacologic and non-pharmacologic interventions, as well as genes or co-morbidities that influence the Parkinson's disease phenotype are extracted from the title of the abstract only. Gene mentions in the title enrich data for those molecular entities that are thought to influence the Parkinson's disease phenotype significantly and are potential therapeutic targets. Bars: Most frequently mentioned 50 of 1862 unique modifiers. Line: Fold change over 5 years, red line = 1. The five most mentioned entities in 2017 are indicated in green boxes; those with highest fold change are shown in white boxes.

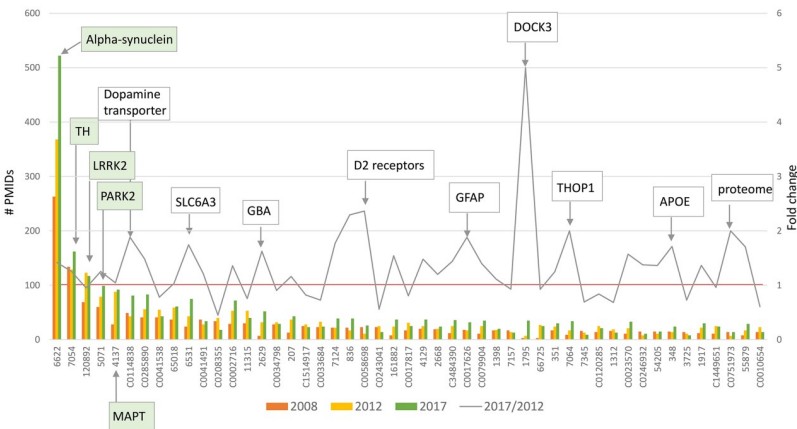

**Fig 3. Number of publications, by gene/molecular concept mention (2008–2017).** Gene mentions in the entire abstract body were extracted to identify a larger group of genes and mechanisms that may not be entities of primary therapeutic interest (extracted in the Interventions module), but that contribute in some way to an understanding of the disease. Bars: Most frequently mentioned 50 of 3311 entities defined by NCBI gene terms or UMLS identifiers (primary data sources). Line: Fold change over 5 years, red line = 1. The five most mentioned entities in 2017 are indicated in green; those with highest fold change in white.

**b. Integrating information across datasets.** The Interventions/Disease Modifier and Genes modules allow the user to rapidly rank these variables by frequency of mention and assess how these fluctuate over time. Next, we integrated additional information regarding species, model, outcome and gene mention data to demonstrate how studies can be aggregated across a diversity of model systems. Two highly mentioned entities extracted by the Interventions/Disease Modifiers module, L-DOPA and alpha synuclein (Fig 2) were selected to illustrate this (Fig 4).

**Patterns of species and model use**: Identified trends were consistent with symptomatic treatment of striatal denervation (L-DOPA associated studies) compared to the mechanistic approaches inherent in understanding the role of alpha-synuclein in PD, and its potential as a therapeutic disease-altering target [77]. In humans, L-DOPA is used as a primary or comparator treatment in humans or animals [78], as well as in human and animal studies of levodopa induced dyskinesia (LID) [79]. Animal studies in which L-DOPA was mentioned relied heavily upon MPTP and 6-OHDA induced models of striatal denervation. Non-human primates are used more frequently in L-DOPA associated studies, consistent with their prominent role in development of treatments for motor symptoms of PD [60]. This trend was consistent with reported literature describing symptomatic models of PD [48]. Studies examining the role of alpha synuclein in animals were dominated by those in mice [67], followed by rats. Because alpha-synuclein is the major component of hallmark Lewy bodies in Parkinson's disease, human reports were also highly represented. Consistent with the literature [61–63, 68] alpha synuclein focused studies were quite well represented in fish, invertebrate models and yeast.

**Patterns of overall outcome and functional outcome reporting**: The proportion of promising or mixed overall outcomes (65% and 27% respectively) was higher for L-DOPA associated studies than for alpha-synuclein associated studies (56% and 16% respectively). Conversely, the proportion of negative outcomes was higher in alpha-synuclein associated studies (14% vs 5%). This reflects a previously reported trend in animal studies in which mechanistic insight is gained through genetic interventions that worsen the phenotype [22]. L-DOPA associated studies reported functional outcome measures at a much higher rate (90%) than those assessing the role of alpha-synuclein (27%). We have noted previously that

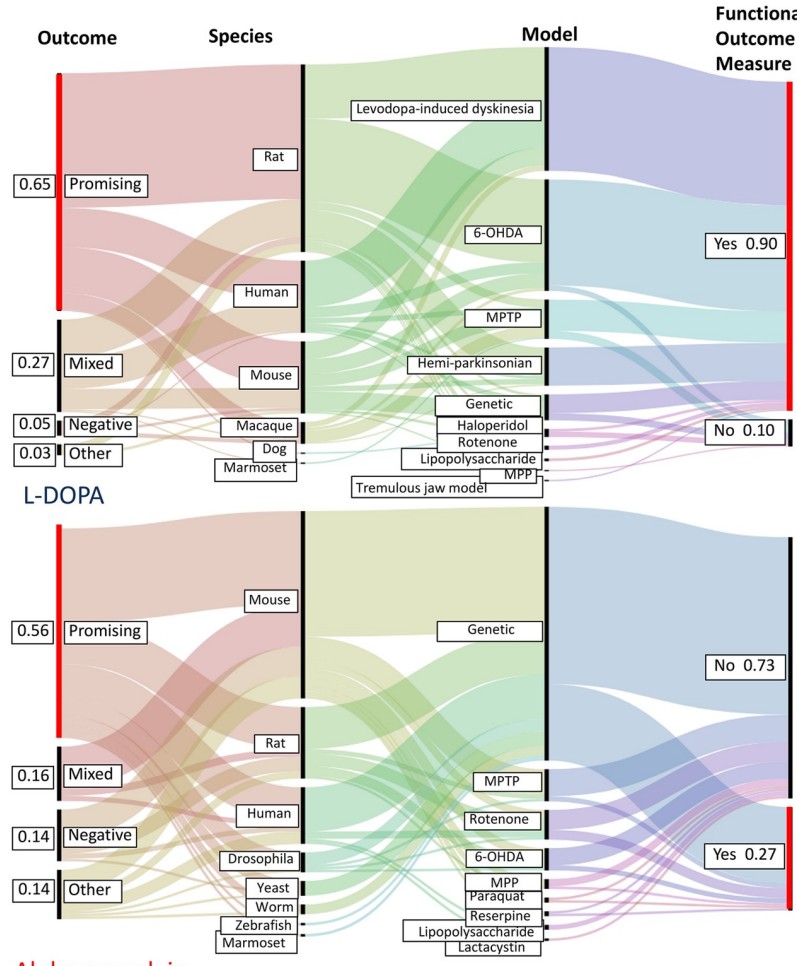

**Fig 4. A schematic comparison of species, model, outcome and functional measure characteristics for an established intervention (L-DOPA) and a target of experimental therapeutic interest (alpha-synuclein).** While both interventions were associated with a majority of promising outcomes, these prevailed in studies mentioning L-DOPA (65% vs 56%). In contrast, the proportion of functional measure reporting was far higher in L-DOPA associated studies (90%) than in alpha-synuclein associated studies (27%). Patterns of species and model use confirmed the expected utility of toxic models in L-DOPA associated studies compared to genetic models in alpha-synuclein associated models. Successive filtering by any of these variables can produce a subset of PMIDs for manual inspection (PMIDs not shown in this schematic). Image produced using RAW graphs (http://app.rawgraphs.io/).

reporting of functional outcome measures appears to associate with those interventions that are approved for use in PD [22].

**Gene mentions by intervention**: Because a user may wish to explore potential molecular pathways associated with a given intervention, we compared mentions of genes (e.g. ERK1) and gene-related terms (e.g. ERK1-2 Pathway) in L-DOPA or alpha synuclein associated abstracts. We extracted 388 and 689 unique entities (defined by NCBI or UMLS identifiers) respectively (S4 Table). All studies in which alpha-synuclein appeared in the title reported genes in the abstract, whereas 130/353 studies with L-DOPA in the title did not, consistent with the often functional rather than molecular nature of L-DOPA associated studies. The most 15 significantly enriched pathways for L-DOPA and alpha-synuclein associated studies (https://reactome.org/; [80]) are shown in S5 Table. As expected, pathways describing dopaminergic, serotonergic and NMDA receptor signaling prevail in L-DOPA associated studies[81],

| Pathway name (L-DOPA associated studies) | # Species | Human | Macaque | Marmoset | Dog | Mouse | Rat | Zebrafish | Fly | Worm |
|---|---|---|---|---|---|---|---|---|---|---|
| Signal Transduction | 5 | ■ | ■ | | | ■ | ■ | | | |
| Signaling by GPCR | 5 | ■ | ■ | | | ■ | ■ | | | |
| Class A/1 (Rhodopsin-like receptors) | 5 | ■ | ■ | | | ■ | ■ | | | |
| GPCR downstream signalling | 5 | ■ | ■ | ■ | | ■ | | | | |
| G alpha (i) signalling events | 5 | ■ | ■ | | | ■ | ■ | | | |
| GPCR ligand binding | 5 | ■ | ■ | | | ■ | ■ | | | |
| Amine ligand-binding receptors | 4 | ■ | ■ | | | ■ | ■ | | | |
| Dopamine receptors | 4 | ■ | ■ | | | ■ | ■ | | | |
| Post NMDA receptor activation events | 4 | ■ | | | | ■ | ■ | | | |
| CREB1 phosphorylation through NMDA receptor-mediated activation of RAS signaling | 4 | ■ | | | | ■ | ■ | | | |
| Neurotransmitter receptors and postsynaptic signal transmission | 3 | ■ | | | | ■ | ■ | | | |
| Transmission across Chemical Synapses | 3 | ■ | | | | ■ | ■ | | | |
| Neuronal System | 3 | ■ | | | | ■ | ■ | | | |
| Signaling by NTRKs | 3 | ■ | | | | ■ | ■ | | | |
| Adenosine P1 receptors | 3 | ■ | | | | ■ | ■ | | | |
| Activation of NMDA receptors and postsynaptic events | 3 | ■ | | | | ■ | ■ | | | |
| MAPK targets/ Nuclear events mediated by MAP kinases | 3 | ■ | | | | ■ | ■ | | | |
| RAF/MAP kinase cascade | 3 | ■ | | | | ■ | ■ | | | |
| Estrogen-dependent nuclear events downstream of ESR-membrane signaling | 3 | ■ | | | | ■ | ■ | | | |
| **Pathway name (alpha synuclein associated studies)** | **# Species** | **Human** | **Macaque** | **Marmoset** | **Dog** | **Mouse** | **Rat** | **Zebrafish** | **Fly** | **Worm** |
| Amyloid fiber formation | 7 | ■ | | ■ | | ■ | ■ | ■ | ■ | ■ |
| FOXO-mediated transcription | 4 | | | | | ■ | ■ | ■ | | ■ |
| CD28 co-stimulation | 3 | | | | | ■ | ■ | ■ | | |
| Mitophagy | 3 | | | | | ■ | ■ | | ■ | |
| Pink/Parkin Mediated Mitophagy | 3 | | | | | ■ | ■ | | ■ | |
| TNFR1-mediated ceramide production | 3 | ■ | | | | ■ | | | ■ | |

**Fig 5. Pathways identified in three or more species for L-DOPA and alpha synuclein associated studies.** Gene lists identified in S4 Table were organized by species, then submitted to the Reactome (https://reactome.org/) to obtain pathways shared by species. Publications associated with an established and approved intervention such as L-DOPA identify pathways that are shared across a greater number of species than those identified by alpha synuclein related studies.

whereas those describing a broad range of cellular events including protein degradation and trafficking, apoptosis and transcriptional control prevail in alpha synuclein associated publications [82].

**Gene mention by species**: Defining the distribution of individual gene mentions across species would provide a starting point to assess how generalizable its associated cellular mechanisms could potentially be to humans. To illustrate this, the species distribution of individual genes co- mentioned with alpha-synuclein associated studies is given in S6 Table. Apart from SNCA itself, other genes are mentioned in 4 or fewer species. Fig 5 illustrates those pathways (https://reactome.org/; [80]) that are shared across 3 or more species in L-DOPA and alpha synuclein associated studies.

**Functional outcome reporting by intervention type**: We noted that the difference in functional outcome reporting between L-DOPA and alpha synuclein related studies was marked. To assess whether the pattern of functional outcomes reporting was limited to these two entities, or extended in a more general fashion to approved or experimental therapeutic approaches, we manually classified entities extracted by the Interventions/Disease Modifiers module as Established or Experimental (S7 Table). Those interventions (and their associated targets, e.g., dopamine receptors) that are already approved in the United States for PD and its complications, or clinically utilized supportive therapies such as exercise or physical therapy, were classified under Established. These were defined according to literature reviews [72, 73, 83] and all achieve symptomatic relief rather than slowing of disease progression. The remainder were classified as Experimental and include a heterogeneous group of terms, including gene terms.

Overall, studies mentioning Established interventions reported a functional outcome measure in 79% of studies, compared to 45% of those studying Experimental interventions (Table 5). When this finding was broken down by species, functional outcome measures were reported across most species used to test Established interventions. A similar finding was noted in far fewer species in which Experimental interventions were co-mentioned. These data are consistent with previous observations[20, 22].

**Table 5. Reporting of functional outcome measure by intervention type.**

|  |  | Functional outcome measure reported | |
| --- | --- | --- | --- |
|  | Total PMIDs | YES | % YES |
| ESTABLISHED | 288 | 227 | 0.79 |
| EXPERIMENTAL | 2945 | 1338 | 0.45 |
| **ESTABLISHED, by species** | **Total PMIDs** | **YES** | **% YES** |
| Human | 67 | 61 | 0.91 |
| Macaque | 13 | 11 | 0.85 |
| Marmoset | 5 | 5 | 1.00 |
| Baboon | 1 | 0 | 0.00 |
| Dog | 1 | 0 | 0.00 |
| Cat | 2 | 1 | 0.50 |
| Rat | 161 | 129 | 0.80 |
| Mouse | 78 | 52 | 0.67 |
| Zebrafish | 1 | 1 | 1.00 |
| Worm | 1 | 1 | 1.00 |
| **EXPERIMENTAL, by species** | **Total PMIDs** | **YES** | **% YES** |
| Human | 1340 | 672 | 0.50 |
| Macaque | 23 | 14 | 0.61 |
| Marmoset | 12 | 10 | 0.83 |
| Vervet monkey | 5 | 2 | 0.40 |
| Baboon | 1 | 0 | 0.00 |
| Dog | 5 | 2 | 0.40 |
| Cat | 20 | 8 | 0.40 |
| Rat | 752 | 358 | 0.48 |
| Mouse | 889 | 339 | 0.38 |
| Rabbit | 3 | 1 | 0.33 |
| Frog | 4 | 4 | 1.00 |
| Zebrafish | 20 | 13 | 0.65 |
| Drosophila | 87 | 34 | 0.39 |
| Worm | 36 | 8 | 0.22 |
| Yeast | 46 | 4 | 0.09 |

## Discussion

Novel discoveries, models and methods using animals constitute a significant portion of federally funded research [16] and are viewed as a means to improve drug discovery rates in the pharmaceutical industry[84]. The Investigative New Drug application (IND) represents a bridge between preclinical and clinical stages, however data in support of an IND focuses primarily on pharmacokinetic/pharmacodynamic data and toxicology, rather than efficacy [85]. Evidence for the latter often has its roots in academic discovery literature, which is heavily weighted toward publication of promising studies in animals [10, 11]. This underscores the need for accurate and realistic assessment of discovery data emerging from academia.

Generalizability, or external validity, is the extent to which research findings derived in one experimental context can be reliably applied to another. Most experimental animal systems tend to be reductionistic (defined as minimizing experimental variables to isolate the phenomenon of interest) [30]. Therefore, attempting to extrapolate from these to complex human systems represents a major translational hurdle [31, 32]. Because common neurodegenerative diseases appear to be a uniquely human phenomenon, it is likely that ideal translational animal

models will not materialize. Instead, identifying which mechanisms or approaches have demonstrated consistent results across different animal systems that reflect aspects of the human disease may have a greater chance of translating to humans [42]. This paradigm is central to the Food and Drug Administration Animal Rule, a mechanism through which a product may be approved when human testing is not feasible for ethical reasons. In this scenario, efficacy must be demonstrated across more than one species using animal study endpoints that are clearly related to the desired outcome in humans [43]. Using this argument as a basis, our program allows the user to assess the diversity of animal (including human) systems across which these have been studied, and to filter these by various criteria, including whether functional measures [20] have been used to determine efficacy. This allows the user to rapidly organize abstract data for an entire disease corpus to yield a smaller subset of papers that can be manually assessed for potential generalizability of the mechanism from animals to humans.

## Comparison with existing resources

Our tool shares some features with those utilized by PubTator [37]. Both our program and PubTator are able to retrieve species (using distinct algorithms) and gene (using the same program GNormPlus). Models, outcomes and reporting of functional outcome measures are unique to our program. A conceptually similar tool is described by Zwierzyna and Overington, 2017[86], in which descriptions of drug screening-related assays in rodents can be screened for mentions of genetic and experimental disease models, treatments, phenotypic readouts and disease indications. Datasets are retrieved from ChEMBL, an open-source manually curated database of bioactive molecules utilized in preclinical drug discovery.

## Novel aspects of our program

PubMed represents a major source of preclinical data that has embedded within it, substantial efficacy data. Our program is uniquely able to organize, filter and compare large scale human and animal model abstract data by translationally relevant criteria. We have demonstrated that we are able to recapitulate expected patterns of mechanistic discovery, species distribution and animal model use in the PD field. The Interventions/Disease Modifier and Genes modules allow the user to rapidly rank individual genes or therapeutic approaches by frequency of mention, and determine how these have changed over time. This approach identifies established approaches, as well as those that are recently emerging. It is this latter group that is of most interest, as accurate assessment of translational potential at this stage could accelerate therapeutic development. Within this group, over-reporting of promising results presents the first hurdle in assessing potential efficacy. Consistent with previous reports [11, 12], our program identified promising outcomes in the majority of studies, confirming that use of this criterion alone is not useful in identifying promising therapies. To overcome this, we are able collect additional species, model and outcome data that can be used to assess potential generalizability of a given gene or therapeutic approach. In agreement with previous observations[20, 22], functional outcome measures were more highly reported in established therapies compared to those that were experimental. Because discovery of useful cellular mechanisms can precede approval of related drugs by decades, testing the hypothesis that reporting of this variable, as well as other patterns extracted by our program, can predict which approaches are likely to generalize successfully to humans will require much larger longitudinal datasets.

## Challenges and limitations

Because animal models are quite specific to various disease fields, our approach was to focus within a specific field (PD) for which we had already identified translational patterns [22]. The

program was able to return expected patterns of species and model use [41]. A limitation of our tool is that we did not define rodent strains, or specific genetic models. This capacity would be useful in discerning the diversity of genetic systems used to explore responses to a given intervention and would be an improvement to consider when extending this tool to fields in which genetic models are primarily used (e.g. Alzheimer's disease).

Data collection for the Interventions/Disease Modifier module is limited to title only. This approach was chosen because we had encountered an unacceptably high false positive rate when the entire abstract was used. A common source of false positive intervention data were gene names. Because we wished to capture genetic interventions central to many rodent studies, we limited collection of gene terms within this module to title only. In this way, our program is biased toward collection of Interventions/Disease Modifier terms, including genetic modifications, that the authors deem worthy of inclusion in the title.

Using the genes module, all gene mentions (and thus associated cellular pathways) in an abstract can be tracked and associated with a given intervention, species or model. This represents an essential step in linking mechanisms of a potential therapy with cellular disease mechanisms. Inherent in both the Interventions/Disease Modifier and Genes module is the need to collapse related terms to a common identifier to support subsequent analyses. In the Genes module, NCBI Gene IDs were annotated in the gold standard, whereas SemRep primarily extracts UMLS Metathesaurus concept identifiers (in addition to, some NCBI Gene identifiers). Second, the NCBI Gene database has different identifiers corresponding to the same gene in different organisms. GNormPlus takes this into account, and tries to extract the appropriate identifier for the species discussed in the abstract, whereas SemRep only considers the identifiers for the Homo Sapiens taxonomy. For the Interventions/Disease Modifier module, manual mapping of outlier terms that did not achieve a match with a MeSH or UMLS identifier was required. This process requires updating as the program is extended to accommodate other neurodegenerative diseases.

## The scope of the program

Study design quality comprises aspects that promote external validity (an example would be use of outcome measures that are shared across humans and animals), and those that promote internal validity/reduce bias (criteria described in the ARRIVE guidelines). Both are important if results are to achieve translation. Our tool is intended to support assessment of external validity or potential generalizability using criteria other than the stated conclusion of efficacy. The tool is not designed to automatically assess well-studied [87] aspects of study design in which full text data is mined to determine whether reporting according to standardized guidelines [24] has occurred. However, the user can very rapidly (and in unbiased fashion) collect and filter the entire published corpus of a disease area to a relevant subset that can then be read to assess measures used to reduce bias, as well as design choices that influence generalizability. The latter are context driven (e.g. the relationship between model choice [34] and mechanism of the intervention [22], or integration of biomarker data with clinically relevant outcome measures) and difficult to accurately address using automatic methods. These, and study design methods aimed at avoiding bias must still be assessed by an individual with implicit knowledge of the field and an understanding of biologic differences across species that influence translation.

Design of the program is modular, and readily adaptable to other disease areas such as Alzheimer's disease. Because neurodegenerative diseases exhibit molecular, clinical and therapeutic overlap, the tool will facilitate integrated evaluation of this group of conditions, and may reveal patterns that result in more efficient animal use.

## Supporting information

**S1 File. Signal lists for species, models, functional outcome measure and interventions with full list of abstracts included in the large-scale evaluation dataset.** Each of these is categorized by Primary (1) or Secondary (2) data type.
(XLSX)

**S1 Table. Proportions of outcome reporting by outcome type in primary and secondary data sources.** Primary data sources included clinical trials and journal articles. Meta-analyses were included in this category as they yield new insights through statistical analysis of multiple studies. Reviews, case reports, editorials and letters comprised secondary data. In general, approximately 70% of studies report a promising outcome (POS) with proportions in primary data sources being slightly lower than in all data sources. In both datasets, a slightly increasing trend in promising reports across 10 years was noted.
(DOCX)

**S2 Table. Interventions/disease modifiers extracted over 10 years.** This module extracts pharmacologic and non-pharmacologic interventions, as well as genes or co-morbidities that influence the Parkinson's disease phenotype. These are extracted from the title of the abstract only.
(DOCX)

**S3 Table. Gene mentions extracted over 10 years.** These are extracted from the entire abstract. Multiple synonyms describing the same entity map to a common UMLS or NCBI identifier.
(DOCX)

**S4 Table. Genes extracted from entire abstract in papers in which L-DOPA or alpha-synuclein were extracted by the interventions module.** Genes are listed by text term and corresponding NCBI or UMLS identifier and are ranked in descending order.
(DOCX)

**S5 Table. The most 15 significantly enriched pathways for L-DOPA and alpha-synuclein associated studies are shown.**
(DOCX)

**S6 Table. Gene mentions across species in papers in which alpha synuclein has been identified using the Interventions module.** The number of studies mentioning each gene by species is colored coded (green = least, red = most).
(DOCX)

**S7 Table. Interventions classified as established.** We manually classified entities extracted by the Interventions/disease modifiers module as Established or Experimental. The former list is presented below—all remaining Interventions were included in the Experimental category.
(DOCX)

## Acknowledgments

The authors gratefully acknowledge the advice and guidance of Thomas Rindflesch (NLM intramural research program), without whose support this work would not have progressed from vision to reality.

## Author Contributions

**Conceptualization:** Caroline J. Zeiss.

**Data curation:** Caroline J. Zeiss, Brent Vander Wyk, Amanda P. Beck, Natalie Zatz, Charles A. Sneiderman.

**Formal analysis:** Caroline J. Zeiss, Dongwook Shin, Brent Vander Wyk, Halil Kilicoglu.

**Funding acquisition:** Caroline J. Zeiss, Halil Kilicoglu.

**Investigation:** Halil Kilicoglu.

**Methodology:** Caroline J. Zeiss, Halil Kilicoglu.

**Project administration:** Caroline J. Zeiss, Halil Kilicoglu.

**Resources:** Halil Kilicoglu.

**Software:** Dongwook Shin, Brent Vander Wyk, Halil Kilicoglu.

**Supervision:** Caroline J. Zeiss.

**Validation:** Dongwook Shin, Halil Kilicoglu.

**Visualization:** Caroline J. Zeiss, Halil Kilicoglu.

**Writing – original draft:** Caroline J. Zeiss, Halil Kilicoglu.

**Writing – review & editing:** Caroline J. Zeiss, Halil Kilicoglu.

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
