## [Decision Letter · Decision Letter 0]

4 Oct 2019

PONE-D-19-18323

Menagerie: A text-mining tool to support animal-human translation in neurodegeneration research

PLOS ONE

Dear Dr. Zeiss,

Thank you for submitting your manuscript to PLOS ONE. After careful consideration, we feel that it has merit but does not fully meet PLOS ONE’s publication criteria as it currently stands. Therefore, we invite you to submit a revised version of the manuscript that addresses all  the points raised during the review process.

We would appreciate receiving your revised manuscript by Nov 18 2019 11:59PM. To enhance the reproducibility of your results, we recommend that if applicable you deposit your laboratory protocols in protocols.io, where a protocol can be assigned its own identifier (DOI) such that it can be cited independently in the future. For instructions see: http://journals.plos.org/plosone/s/submission-guidelines#loc-laboratory-protocols

We look forward to receiving your revised manuscript.

Kind regards,

Gianluigi Forloni

Academic Editor

PLOS ONE

Journal Requirements:

1. Thank you for including your competing interests statement; "No"

Reviewers' comments:

Reviewer's Responses to Questions

**Comments to the Author**

1. Is the manuscript technically sound, and do the data support the conclusions?

Reviewer #1: Yes

2. Has the statistical analysis been performed appropriately and rigorously? 

Reviewer #1: I Don't Know

3. Have the authors made all data underlying the findings in their manuscript fully available?

Reviewer #1: Yes

4. Is the manuscript presented in an intelligible fashion and written in standard English?

Reviewer #1: Yes

5. Review Comments to the Author

Reviewer #1: PONE-D-19-18323

This paper is well written, but it is long and complex. Some editing and clarification, especially in the methodology and results sections, would be helpful. The results section seems to contain some elements that should be within the methodology section. I found the Results section difficult to understand. The key findings need to be brought out; for example it seems that reporting of functional outcomes is a key predictor of study efficacy, but this wasn’t emphasised (or perhaps I have misunderstood).

The authors aim to improve external validity by aggregating evidence across a range of models, arguing that single animal models are unable to recapitulate human neurodegenerative diseases. They argue that consistent results across diverse animal models may clarify whether an approach has translational potential. I think they are correct in highlighting the problems of animal-human translation and in focusing on efficacy. Moreover, text mining approaches have the advantage of being able to offer insights about animal models without using further animals. However, I’m unclear whether their approach will improve external validity, as they claim. They note (pp36-37) that translation is difficult because animal models are reductionist, but suggest this can be overcome by identifying mechanisms that show consistent results across diverse animal models that reflect different aspects of the human disease, noting that FDA requires testing in more than one species. The problem is, this does not ensure external validity. Even if some aspects of the human disease are reflected in diverse animal models, not all aspects of the human disease are going to be accurately modelled. Furthermore, external validity is always going to be compromised because of the problem of animal-human species differences. Although mice and humans have genetic, biochemical and physiological similarities, our lineages diverged around 85 million years ago and since then, have become adapted to very different environments. Mice and other animals are unlikely to be useful for understanding modern, non-communicable diseases in humans, because the causes of these arise from our unique, evolved life histories. I think it would help if the authors acknowledged this.

P3, line 20: needs to expand on the reasons for translational failure, i.e. not just state that they are complex but tease out the different factors that impact translation, i.e. publication bias, poor reporting, poor study design, poor conduct of studies etc. (some of which are considered on p4). This will help contextualise the study so that when the authors state that they are addressing external validity, readers will understand where this fits into the translational picture.

P3, lines 22-23: this isn’t quite true – the discovery literature is heavily weighted towards publication of studies that claim to be promising, but that if took measures to prevent bias would not achieve such positive results, or that if set in the context of published negative studies would not skew entire fields towards excess significance.

P4, line 1: is dichotomy the right word?

Line 9, page 5: too many ‘are’s.

P41, lines 6-7: Incorrect definition of publication bias in brackets. The tool doesn’t allow users to overcome the problem of publication bias (after all, it cannot take into account unpublished negative studies); rather, it allows users to ignore authors’ claims about their studies’ ‘promising’ results, which is a different phenomenon.

Is the tool compromised by being unable to take study quality into account?

6. PLOS authors have the option to publish the peer review history of their article (what does this mean?). If published, this will include your full peer review and any attached files.

Reviewer #1: No

---

## [Author Response · Author response to Decision Letter 0]

12 Nov 2019

Nov 11, 2019

We thank the reviewers for their thoughtful comments which we feel have improved readability and helped clarify several issues for the reader. All have been addressed as described below. Reviewers comments are underlined, followed by our responses.

The authors have confirmed that no competing interests exist

2. Has the statistical analysis been performed appropriately and rigorously? 

Reviewer #1: I Don't Know

The paper does not contain statistical analyses. Intrinsic evaluation was performed using standard information extraction evaluation metrics: precision (or positive predictive value), recall (sensitivity), and F1 score (the harmonic mean of precision or recall). All tables and graphs shown for large scale evaluation are descriptive data (predominantly describing proportions).

 5. Review Comments to the Author

Reviewer #1: PONE-D-19-18323

1. This paper is well written, but it is long and complex. Some editing and clarification, especially in the methodology and results sections, would be helpful. The results section seems to contain some elements that should be within the methodology section. I found the Results section difficult to understand. 

The manuscript has been shortened (currently 8973 words vs the original 9783 words (incl abstract, summary, tables and figure legends) and edited as follows: 

a. The introduction has been edited to address points raised in point 4 below.

b. The methods section has been left intact, as removing text here would remove content, and we feel this is a critical portion of the paper. However, some material from the results section has been moved here. 

c. The results section has been shortened and edited for clarity. The various types of comparisons that could conceivably go in the methods section have been left here to avoid having to repeat these in the results section. The following sentence has been added to clarify this: Page 18, line 4:” Specific comparisons are given by subheading in the results section.” 

d. The final portion (outcomes by publication type) has been moved to the beginning, with subsequent elimination of one Supp Table through combining the original S1 Table and S8 Table into a new S1 Table. 

e. The discussion section has been substantially shortened, primarily through removal of redundant text already mentioned in the methods or results section. Additionally, subheading that highlight main points are used. A subheading to clarify limitations and challenges has been used to address the last point raised by the reviewer. 

2. The key findings need to be brought out; for example it seems that reporting of functional outcomes is a key predictor of study efficacy, but this wasn’t emphasised (or perhaps I have misunderstood).

It appears from this study and a previous one (Zeiss et al, 2017) that use of functional outcome measures do indeed have the potential to predict efficacy that translates to humans. We are careful to mention that actually proving this point requires confirmation in a larger dataset as follows

Page 34, line 11-17: “In agreement with previous observations [26, 28], functional outcome measures were more highly reported in established therapies compared to those that were experimental. Because discovery of useful cellular mechanisms can precede approval of related drugs by decades, testing the hypothesis that reporting of this variable, as well as other patterns extracted by our program, can predict which approaches are likely to generalize successfully to humans will require much larger longitudinal datasets” 

Zeiss CJ, Allore HG, Beck AP. Established patterns of animal study design undermine translation of disease-modifying therapies for Parkinson's disease. PLoS One. 2017 Feb 9;12(2):e0171790.

(PS The effort described above has just started using a 80K-abstract PD dataset spanning 1950-2019)

3. The authors aim to improve external validity by aggregating evidence across a range of models, arguing that single animal models are unable to recapitulate human neurodegenerative diseases. They argue that consistent results across diverse animal models may clarify whether an approach has translational potential. I think they are correct in highlighting the problems of animal-human translation and in focusing on efficacy. Moreover, text mining approaches have the advantage of being able to offer insights about animal models without using further animals. However, I’m unclear whether their approach will improve external validity, as they claim. They note (pp36-37) that translation is difficult because animal models are reductionist, but suggest this can be overcome by identifying mechanisms that show consistent results across diverse animal models that reflect different aspects of the human disease, noting that FDA requires testing in more than one species. The problem is, this does not ensure external validity. Even if some aspects of the human disease are reflected in diverse animal models, not all aspects of the human disease are going to be accurately modelled. Furthermore, external validity is always going to be compromised because of the problem of animal-human species differences. Although mice and humans have genetic, biochemical and physiological similarities, our lineages diverged around 85 million years ago and since then, have become adapted to very different environments. Mice and other animals are unlikely to be useful for understanding modern, non-communicable diseases in humans, because the causes of these arise from our unique, evolved life histories. I think it would help if the authors acknowledged this.

The fundamental differences between human and animal cell biology that can derail translation are extremely important and have now been alluded to in the introduction and discussion. 

Page 5, line 6-7: “Profound differences in animal and human physiology[12] are a critical source of poor translation”

Page 36, line 20-21:” These, and study design methods aimed at avoiding bias must still be assessed by an individual with implicit knowledge of the field and an understanding of biologic differences across species that influence translation”.

There are many days when I share the reviewer’s view on the poor chances of mice/other animals to understand complex diseases in humans. However, the issue is extraordinarily complex. For example, key steps in the use of LDOPA in PD involved animals. These included aspects of catecholamine biochemistry (PMID: 16746563), the universality of catecholamine distribution in the brain (PMID: 13451690); depletion of these stores by reserpine in the rabbit (PMID: 13552704) and restoration of resulting functional deficits by dihydroxyphenylalanine in reserpinized mice (PMID: 13483658) prior to treatment in PD patients (PMID: 13869404). Modulation of striatal neurochemistry by adenosine A2A receptors was demonstrated in rats (PMID: 7812630, 8780014) almost 2 decades before istradefylline’s first approval in Japan.

We fully recognize that these drugs achieve symptomatic relief through a highly conserved nigrostriatal circuitry, and that the molecular underpinnings of dopaminergic cell death are more difficult to address. It is very difficult to know which discoveries eventually have clinical application. Given the financial and intellectual resources directed at animal research, the capacity to assess the generalizability of interventions across different systems in an unbiased manner may be useful. 

We have made it clear that what is presented here is the program design and accuracy as evidenced by its ability to extract known patterns from PD literature and that testing whether the program can predict translational success will require its application to a much larger dataset. 

4. P3, line 20: needs to expand on the reasons for translational failure, i.e. not just state that they are complex but tease out the different factors that impact translation, i.e. publication bias, poor reporting, poor study design, poor conduct of studies etc. (some of which are considered on p4). This will help contextualise the study so that when the authors state that they are addressing external validity, readers will understand where this fits into the translational picture.

The introduction has been edited to clarify aspects of internal and external validity in animal study design that together impede translation (Page 4 and 5). It has been made clear that the intent of the program is to address some aspects of external validity such as diversity of model systems and use of functional outcome measures, and that what is presented here is the program design and accuracy as evidenced by its ability to extract known patterns from PD literature. 

P3, lines 22-23: this isn’t quite true – the discovery literature is heavily weighted towards publication of studies that claim to be promising, but that if took measures to prevent bias would not achieve such positive results, or that if set in the context of published negative studies would not skew entire fields towards excess significance.

This sentence has been followed in the next paragraph by an explanation of factors that skew published animal research in the direction of excess significance, as well as a summary of measures the research community is /should be taking to address this, including publication of negative results. 

Page 4, lines 10-19:” Several factors that undermine the reliability of animal studies have been identified. These include insufficient rigor in animal study design and reporting [6, 23], publication bias [16, 18], over-reporting of significance [24, 25], over-reliance on non-clinical outcome measures [26], and entrenched use of certain model systems [3, 27, 28]. Together, these issues contribute to poor reproducibility of animal studies, and certainly worsen the translational gap[29]. To address this, reporting and design guidelines [9] have been adopted by numerous journals[10], and by major funding agencies[11]. These, once implemented, should improve rigor and reproducibility of preclinical studies, although widespread evidence for this is not yet available [30]. Regarding publication bias, greater reporting of negative studies [31, 32] would provide a more realistic view of actual preclinical efficacy. “

P4, line 1: is dichotomy the right word?

Dichotomy has been replaced with “this translational gap”

Line 9, page 5: too many ‘are’s.

Thank you “are” has been replaced with “and” 

P41, lines 6-7: Incorrect definition of publication bias in brackets. The tool doesn’t allow users to overcome the problem of publication bias (after all, it cannot take into account unpublished negative studies); rather, it allows users to ignore authors’ claims about their studies’ ‘promising’ results, which is a different phenomenon.

Yes, this is a good point. The sentence referring to publication bias has been removed and replaced with the following:

Page 36, lines 8-9: “Our tool is intended to support assessment of external validity using criteria other than the stated conclusion of efficacy. “ 

Is the tool compromised by being unable to take study quality into account?

We agree that study quality is crucial, and must be evaluated. Design quality comprises aspects that promote external validity (an example would be use of outcome measures that are shared across humans and animals, use of a progressive chronic model if neuroprotective therapies are studied or replication across different species), and those that promote internal validity/reduce bias (e.g reporting criteria in the ARRIVE guidelines).

Our program is designed to support assessing the former using abstract data on a large scale, and builds upon concepts introduced in our previous publications (PMID: 25448761; 28182759). 

In a previous publication (PMID: 28182759), we compared reporting of various criteria in support of external and internal validity, and found that entities such as animal number, statistical tests used, blinding, housing etc were variably reported even for interventions that went on to be approved. These measures were reported in various locations (methods, figure legends, Supp data) of the full text article, thus complicating automatic analysis. In contrast, features supporting external validity (e.g species and model diversity, and use of functional measures) were reported more frequently in approved interventions, and are often mentioned in abstracts, and were thus chosen as aspects of our program. We feel that because studies are often complex, and contextual, automatic exclusion based on failure to report a given ARRIVE guideline could exclude useful papers. Further, it would require us to develop an entirely different program that searches full text papers that are more publicly restricted than abstracts). Having a human with implicit knowledge of the field is a critical aspect of final decision making on study quality – our program very rapidly filters numbers of studies (PMID: 25448761) to a readily manageable subset for manual inspection of design factors that support both internal and external validity. 

While ARRIVE guidelines have not been broadly implemented yet (PMID: 29795636) there is at least growing awareness of the need to design studies to avoid bias. There is far less awareness of design factors needed to promote generalizability - our program is focused on that problem. 

We have made the abilities and limits of the program clear to the reader in the last part of the discussion: page 36-37, under the subheading: Scope of the program.

---

## [Decision Letter · Decision Letter 1]

22 Nov 2019

Menagerie: A text-mining tool to support animal-human translation in neurodegeneration research

PONE-D-19-18323R1

Dear Dr. Zeiss,

We are pleased to inform you that your manuscript has been judged scientifically suitable for publication and will be formally accepted for publication once it complies with all outstanding technical requirements.

With kind regards,

Gianluigi Forloni

Academic Editor

PLOS ONE

Additional Editor Comments (optional):

Reviewers' comments:

Reviewer's Responses to Questions

**Comments to the Author**

1. If the authors have adequately addressed your comments raised in a previous round of review and you feel that this manuscript is now acceptable for publication, you may indicate that here to bypass the “Comments to the Author” section, enter your conflict of interest statement in the “Confidential to Editor” section, and submit your "Accept" recommendation.

Reviewer #1: (No Response)

2. Is the manuscript technically sound, and do the data support the conclusions?

Reviewer #1: Yes

3. Has the statistical analysis been performed appropriately and rigorously? 

Reviewer #1: N/A

4. Have the authors made all data underlying the findings in their manuscript fully available?

Reviewer #1: Yes

5. Is the manuscript presented in an intelligible fashion and written in standard English?

Reviewer #1: Yes

6. Review Comments to the Author

Reviewer #1: (No Response)

7. PLOS authors have the option to publish the peer review history of their article (what does this mean?). If published, this will include your full peer review and any attached files.

Reviewer #1: No

---

## [Editor Report · Acceptance letter]

9 Dec 2019

PONE-D-19-18323R1 

Menagerie: A text-mining tool to support animal-human translation in neurodegeneration research 

Dear Dr. Zeiss:

I am pleased to inform you that your manuscript has been deemed suitable for publication in PLOS ONE. Congratulations! Your manuscript is now with our production department. 

With kind regards,

on behalf of

Dr. Gianluigi Forloni 

Academic Editor

PLOS ONE